# Who gets sicker and why? Parents' perceptions of COVID-19 disparities and how they would explain them to their children

**Lester A. Mejia Gomez**[1], **David Menendez**[2*], **Valerie Umscheid**[1], **Susan A. Gelman**[1]

**1** Department of Psychology, University of Michigan, Ann Arbor, Michigan, United States of America,
**2** Psychology Department, University of California, Santa Cruz, California, United States of America

* damenend@ucsc.edu

## Abstract

The COVID-19 pandemic revealed substantial health disparities, disproportionately impacting Black individuals, individuals of lower socioeconomic status, and older adults in the US. Little is known as to whether and how adults discuss these disparities with their children, an essential first step toward determining when and how children come to understand these differences. To address these questions, we recruited parents with at least one child aged 5–12 ($N = 443$, 61% White) from CloudResearch Prime Panels. We asked participants to report their likelihood of discussing these disparities with their children, how they would explain them, their own beliefs regarding these disparities, and a series of group perception and attitudinal measures. An ordinal mixed-effects regression revealed that parents were significantly more likely to say they would discuss the age disparity than the race and class disparities, with no difference between the latter. Parents of older children reported being more likely to discuss race and age disparities than parents of younger children. Ordinal logistic regressions revealed that parents reported they would discuss the race disparity significantly more when they held stronger racial essentialist beliefs, held stronger racial social constructionist beliefs, and perceived Black people as warmer and less competent. Parents also reported that they would discuss the social class disparity significantly more when they held stronger essentialist beliefs about social class. Qualitative coding revealed that parents' potential explanations for the disparity and reasons to discuss the disparities (or not) with their children differed by dimension. Finally, parents' own beliefs about the existence, nature, and causes of these disparities predicted the likelihood that they would discuss them with their children -- though differently for the different dimensions. Overall, our findings suggest that parents' likelihood of discussing health disparities reflects three key factors: their own beliefs about whether/how such disparities exist, their attitudes toward the affected groups, and their comfort in discussing social issues with their children.

**Data availability statement:** Data is available from https://osf.io/jv9at/.

**Funding:** This research was supported by NSF Grant #2055164 to SAG. Any opinions, findings, and conclusions or recommendations expressed in this material are those of the authors and do not necessarily reflect the views of the National Science Foundation.

**Competing interests:** The authors have declared that no competing interests exist.

## Introduction

Health disparities in the United States have been documented for decades, with minority groups typically enduring the most adverse consequences [1]. These same disparities were evident during the coronavirus (COVID-19) pandemic, with older adults, Black people, and lower-income individuals more likely to be hospitalized due to COVID-19 than younger adults, White people, and higher-income individuals, respectively [2–4]. Although these statistics illustrate a clear disproportionate impact of COVID-19 on these communities, the causes of these disparities are open to multiple interpretations, with important implications for how people reason about differences between social groups. This interpretive ambiguity stresses the need for research investigating how people think about health disparities and communicate them to others. It is especially critical to learn how these ideas are communicated to children, as socialization can predict how children regulate their reasoning and actions and influence their values and attitudes [5].

However, little is known regarding whether these disparities are discussed with children and, if so, how they are explained. The COVID-19 lockdowns in 2020 created a unique household experience, where many parents and their children were confined to their homes for a prolonged period, with some parents shifting to working remotely and children attending school virtually. Although parents are already young children's most active socialization agents, the lockdowns increased the time parents and children spent together. With little to no contact with educators or non-familial caretakers, parents were often the primary source of information to children about the pandemic. Indeed, during the earlier stages of the pandemic, children turned to their parents with questions about the virus [6–8]. For example, parents self-reported that their children's questions (3–12 years old) primarily focused on the consequences of the disease (e.g., "Will mommy and daddy die?" [6] p. 58) and lifestyle changes (e.g., "Why can't we leave the house?" [8] p. 7). In turn, parents reported discussing the COVID-19 virus with their children, primarily focusing on how to prevent it, and what COVID-19 is [9]. However, these studies did not specifically examine how parents and children discuss health disparities.

The current study examines parents' self-reported beliefs regarding group differences in the likelihood of a person getting sick from COVID-19, whether and how they would communicate such disparities to their children, and how these patterns relate to their attitudes and beliefs about the social categories in question.

### Perceptions of health disparities

Gollust and colleagues [10] conducted three waves of data collection in April 2020, August 2020, and April 2021, using a nationally representative sample to gain insight into public opinions of the existing COVID-19 health disparities. They found that in each wave, over 75% of participants agreed that individuals with chronic health issues (vs. those without any) and older adults (vs. younger adults) were more likely to die from COVID-19 complications. However, less than 55% of participants in each wave agreed that Black (vs. White) and poorer (vs. wealthier) people were more

likely to die from COVID-19 complications. The authors note important Black-White racial differences, with White participants agreeing with the chronic health and age disparity most in each wave, Black participants agreeing with the race disparity most during the second and third wave, but no differences in agreement for the social class disparity [10,11].

Menendez and colleagues [12] assessed children's (ages 5–12) and their parents' beliefs about who was more likely to contract COVID-19 across several social comparisons, in a primarily White sample. Whereas most children and parents reported that an older person was more likely to get sick than a young adult, the majority believed that a Black and White person were equally likely to get sick, and just over half believed that a poor and rich person were equally likely to get sick. Parents were more likely to provide biological explanations (i.e., focusing on the body, pre-existing conditions, or immune functions) when choosing an older adult, structural explanations (i.e., focusing on external social factors such as access to healthcare) when choosing a poor person, and behavioral (i.e., focusing on an individual's behavior such as choosing to not wear a mask) and structural explanations when choosing a Black person [12]. This shows that parents have some understanding of health disparities, and they attribute these disparities to different factors depending on the category.

### Parental input regarding race, social class, and age

Parent-child conversations communicate--both explicitly and implicitly--important information that may contribute to a child's beliefs and understanding [5,13,14]. For example, Finch and colleagues [15] found common themes in how parents reported (online) explaining COVID-19 to their children early in the pandemic (May-July 2020) and how their children explained COVID-19 during in-person interviews 6–18 months later. Both the frequency and content of parent-child conversations can play an important role in children's acquisition of attitudes and beliefs about social differences between racial groups [16–18]. At the same time, we do not assume that children simply or automatically adopt their parents' beliefs and attitudes [19]. For example, a meta-analysis found that parent-child racial attitudes share only a low-to-moderate relation, with younger children's attitudes being least similar to those of their parents [20]. Thus, parent-child conversations are but a start, leaving open the further question of whether and how such messages affect children's beliefs or attitudes.

The research cited above did not examine how parents discuss race, age, or social class disparities with their children. However, prior research indicates that White parents' race-related conversations with their children lack quantity, quality, and impact [16,17,21–24]. For instance, Abaied and colleagues [16] explored the nature of these conversations among White families at the height of the 2020 Black Lives Matter protests against police brutality after the murder of George Floyd. They found that many parents did not feel responsible for engaging in these conversations with their children, did not have comprehensive discussions about racism and White privilege using a critical perspective, and often defaulted to colorblind ideologies ("race is not important," p. 9) or challenged "the legitimacy of concerns about racism" (p. 15) altogether.

To our knowledge, little research has examined how parents discuss aging or death with their children, and the existing literature is limited to specific populations, such as children with terminal illnesses [25]. Similarly, there is surprisingly little work on how parents socialize children about social class [26] or on how people explain social class disparities in health outcomes. One study (not focused on parental input to children) examined how adults in a set of focus groups explained health differences between lower- and higher-status individuals in Scotland and Northern England [27]. They found that lower-status individuals often attributed health disparities to structural factors (e.g., lack of access to resources). In contrast, higher-class individuals were more likely to attribute poorer health to individual behavioral reasons (e.g., alcohol use) and resisted the notion that a person's social status affects health. Interestingly, some responses among the lower-SES participants resembled essentialist reasoning ("Or you're marked, as soon as you are born… That's your station in life, don't move out of it," p. 2178), suggesting that some individuals may attribute their experiences to underlying traits about their social group. This limited research highlights the need to investigate whether and how parents discuss

social class health disparities with their children, especially when considering that over 11 million children in the US live under the federal poverty line [28].

In the present study, we focused on disparities linked to age, race, and social class, as these are well-documented with regard to COVID-19 [2–4] and are social categories that young children are aware of from an early age. By 4 years of age, children use a person's age to predict other features of the individual [29], and by preschool age they have acquired stereotypes about older adults [30,31]. Preschool children have also learned the racial categories of their culture and make inferences from these categories [32–34], although there are also important developmental changes in their understanding [35]. Young children are also sensitive to differences based on social class. For example, some children as young as four years old associate wealth with White children but not Black children [36,37], and children as young as 5 and 6 years old have some understanding of what it means to be 'rich' and 'poor' [38]. Finally, recent work also suggests that children are not only aware of social inequalities but may also actively rectify them [39–41].

## Attitudes and beliefs about social categories

One goal of this project was to examine how individual differences in parental attitudes and beliefs might relate to their conversations with their children. Given that there is little prior work examining how parents and children communicate about health disparities, we took an exploratory approach. We selected five individual difference measures that have been found to relate to how people think about social categories and/or parenting behaviors associated with conversations about social categories.

**Stereotype content.** The Stereotype Content Model (SCM) relates group stereotype content to elicited emotions [42] and behaviors [43] based on a society's social structures. The model distinguishes two perceived dimensions: warmth and competence [42–44]. For example, rich people are stereotypically perceived as being low in warmth but high in competence, whereas poor individuals are stereotypically perceived as being low in both warmth and competence [42,43,45,46]. The stereotype content individuals hold about a group is theorized to predict behaviors they may engage in toward that group [44,47,48]. For example, warmth perceptions predict active behaviors, such as either helping or harassment, and competence perceptions predict passive behaviors, such as neglect [43]. In the current study, the extent to which a group is perceived as warm and competent may predict an individual's likelihood of acknowledging and discussing a disparity.

**Internal motivation to respond without prejudice.** The internal and external motivations to respond without prejudice scales assess a person's internal values and beliefs that reduce the likelihood of eliciting prejudiced responses (internal motivation scale; IMS), and the broader social demands that may result in them not wanting to appear prejudiced to avoid social sanctions (external motivation scale; EMS) [49]. Theoretically, a person who is highly motivated to respond without prejudice may be more likely to engage in behaviors that reduce their likelihood of appearing prejudiced. For example, a person may seek information about how ageist stereotypes affect older adults, the core motivations of the Black Lives Matter movement, or how redlining affects lower socioeconomic status communities. IMS and EMS have been found to relate to White people's perceptions of Black people [50], their interactions with Black people [51], and their engagement with reducing or hiding their prejudice [52].

Recent research has also examined how IMS may affect parents' racial socialization strategies [53–55]. For example, Perry and colleagues [54] found that White parents who reported higher IMS were more likely to discuss race and racism with their children. However, there is less evidence that EMS predicts racial socialization [55]. Based on these findings, we focused on IMS, reasoning that IMS could predict parents' likelihood of discussing racial disparities with their children. Similarly, we extended IMS to social class and age disparities to examine if IMS relates to how parents discuss non-racial disparities.

**Category disregard.** A third factor that may contribute to parents' likelihood of engaging in conversations about social issues is the degree to which they deny the role of a person's group membership (e.g., race, social class) in their

experiences in society, also known as category disregard [56]. Theoretical arguments suggest that disregarding a social category could predict whether parents discuss health disparities with their children. Someone who disregards race could believe that one reason for racial health disparities is a group's behaviors (thereby implicitly blaming members of the group) instead of acknowledging systemic racism in society (e.g., underfunded community health centers) [57]. Indeed, empirical evidence has found that disregarding race predicts how parents address racial topics with their children [16,58].

**Essentialism.** Psychological essentialism is the assumption that members of a category possess an inherent, innate, and unmalleable "essence" that constitutes their membership in that category [59–61]. Essentialism has been identified as contributing to harmful stereotypes and attitudes of many social groups [33,62–70]. For instance, individuals who perceive Black-White racial differences as biologically based tend to be more inclined to accept racial inequalities and less interested in interacting with people from another racial group [69]. Psychological essentialism may encourage inaccurate depictions of how viral transmission works (e.g., the erroneous belief that some groups are innately more susceptible to viruses) and why disparities exist across social groups (e.g., the erroneous belief that it is *natural* for some groups to experience inequality).

**Social constructionism.** Conversely, social constructionism refers to the extent to which social categories and their consequences (e.g., racism) are viewed as socio-culturally constructed [71–73]. Kung et al. [71] found that people who endorsed more social constructionist beliefs fostered better interracial trust with others than those who endorsed more essentialist beliefs. Therefore, communicating that race is a social construct and more indicative of a society's view of race can produce better intergroup outcomes than believing that race is a natural kind that determines a person's characteristics and is unchangeable.

In the current study, we examined whether individual differences in parental endorsement of essentialist or social constructionist reasoning are associated with different levels or types of conversations about social group disparities with their children. For example, parents with robust essentialist perspectives may explain group disparities as natural and inevitable, whereas parents with robust social constructionist perspectives may explain group disparities as stemming from systemic factors (e.g., that poorer individuals are disproportionately affected by COVID-19 because they are unable to work from home or have less access to quality healthcare).

## The current study

The present study was designed to examine parental beliefs about age-, race-, and class-related COVID-19 health disparities, parents' likelihood of saying they would be to discuss these disparities with their children, the content of such discussions, and how these conversations relate to parents' own beliefs and attitudes. As a first step toward examining these issues, we employed a parent self-report measure, a method that has been used in other recent research in this area [8,74–76]. Specifically, parents were asked whether they would discuss with their child(ren) why some groups are more affected by COVID-19 than others, and the content or messages they would share with their children if prompted. We focused on parents of 5- to 12-year-old children as this is an important period in the development of reasoning about social categories [77], COVID-19 [9,12], and contagious illnesses more generally [78–80].

The research reviewed above suggests possible patterns in whether and how parents would discuss race disparities with their children [16,17,23,40,54,58,81]. By contrast, parent-child conversations about age and social class [26] have been less explored. Additionally, several factors may play a role in parents' likelihood of engaging in these conversations, including their ideologies [16,58], beliefs [68,69], and motivations [53–55,82]. Given the exploratory nature of this research, we did not preregister any specific hypothesis. However, we address the following research questions:

1) What is parents' likelihood of saying they would discuss COVID-19 health disparities with their children? Does their likelihood depend on their child's age? Does their likelihood depend on whether the disparity is about age, race, or social class?

2) Do parents think that social categories influence the likelihood of someone getting very sick with COVID-19? How do parents reason about these disparities?

3) How do parents say they would explain health disparities to their children?

4) Do parents' intergroup attitudes and beliefs influence their likelihood of saying they would discuss each disparity with their children?

The research questions, analysis plans, and exclusion criteria were preregistered in AsPredicted #115861.

## Materials and methods

### Participants

We preregistered that we would recruit at least 400 participants, based on power analysis in G*Power [83] using linear regression (F test, r2 increase, total predictors: 15, tested predictors: 1, power value: 80%, and alpha value:.05), indicating that a total of 400 participants would enable us to detect effect sizes of $d = 0.04$, the smallest effect size of interest. Participants were recruited using CloudResearch Prime Panels, a crowdsourcing platform that recruits participants from several online research panels [84]. We included parents with at least one child between the ages of 5 and 12 ($N = 507$). The study was approved by the Institutional Review Board of the University of Michigan under HUM00227405. The data were collected in December 2022 and January 2023.

Sixty-four participants were excluded from data analysis due to prescreening (e.g., indicating they were a parent to qualify for the survey on CloudResearch, but later responding they were not parents during the second prescreening in the survey) or failing an attention/bot check (more details on these below). The final sample consisted of parents living in the US ($N = 443$, $M_{age} = 37.04$, $SD = 8.39$), with at least one participant from each of 46 states and the District of Columbia. The three most common states were New York ($n = 36$), Texas ($n = 36$), and Florida ($n = 32$), and the states not represented were Hawaii, Montana, Vermont, and Wyoming. All demographics were self-reported. The gender distribution was 62% women, 33% men, and 5% not reported. Ethnic-racial identity was: 61% White, 20% Black or African American, 5% Hispanic or Latinx, 3% Asian or Asian American, 10% multiple races and/or ethnicities, and 1% other (not reported). Educational attainment was as follows: 2% completed less than high school, 28% graduated high school, 19% completed some college, 12% obtained a 2-year degree, 22% obtained a 4-year degree, 14% obtained a master's degree, and 3% obtained a doctorate degree. The political party affiliations were 44% Democrat, 28% Republican, 24% Independent, and 4% other or no affiliation. Political leaning overall was neither conservative nor liberal ($M = 3.87$, $SD = 1.81$; scale range 1–7, where 1 = very conservative, 7 = very liberal).

### Measures and procedure

Participants self-selected to participate in the current study via CloudResearch Panels. They were then redirected to the Qualtrics survey, where they: verified that they met the study's inclusion criterion (i.e., currently a parent of a child between the ages 5–12); read and digitally approved the informed consent form; and provided reCAPTCHA verification to distinguish themselves from online bots.

After providing consent, participants read, "Do you think some kinds of people are more likely than others to get sick with COVID?" If they responded yes, they were asked, "Who is more likely to get sick with COVID?" and could specify. These responses are not discussed here.

Next, participants received a **Disparity Judgment (Forced-Choice) Task**, consisting of four trials. On each, participants saw images of two characters, one on the left and one on the right, each representing a social group (e.g., Black people vs. White people). Participants were asked, "On average, who's more likely to get very, very sick with COVID?" and could select one of three named choices, consisting of either of the two social groups displayed or no difference (e.g.,

"Black people", "White people", or "They are the same"). We followed with a **Confidence Rating** ("How sure are you?"), using a 4-point Likert scale (1 = not sure, 4 = extremely sure), to include a more nuanced assessment of how strongly parents felt about their previous categorical, forced-choice judgment. There were four blocks, one for each of the following comparisons: old adults vs. young adults, Black people vs. White people, poor people vs. rich people, nice people vs. mean people. The order of blocks and character placement were randomized. We adopted this task from Menendez and colleagues [12].

On each trial, following their confidence rating, participants received an **Explanation Endorsement Task**, where they were asked to explain their answer (either "Why are [group selected] more likely to get very, very sick with COVID?", or "Why are [group 1] and [group 2] just as likely to get very, very sick with COVID?", depending on their prior choice) by agreeing or disagreeing with each of the following reasons: behavioral ("Is it because of how they act or things they do?"), biological ("Is it because of what their bodies are like inside?"), structural ("Is it because of where they live or what they can buy?"), or other ("Is it because of a different reason? Please explain", with an option to specify). The first three choices were listed randomly, and the 'other' choice was always last. The three specified reasons were derived from Menendez and colleagues [12], as these were the most common explanations for COVID-19 disparities among children and adults in response to open-ended questions. Participants then received two tasks that will not be reported here, regarding whether social relatedness (e.g., family member vs. stranger) affects the chances of someone getting sick or the likelihood of engaging in various health protective behaviors.

Participants received an attention/bot check midway through the survey, where they were asked to name or describe the image on their screen (a COVID-19 virus) in an open-ended textbox. Participants who did not describe the image or did not provide a reasonable response were allowed to complete the survey but were excluded from the analyses.

After these tasks, participants completed a **Disparity Judgment (Scale) Task**, one for each comparison. This was designed to provide a more fine-grained and potentially more sensitive measure than the forced-choice task. They read, "16 people went to a center to get tested for COVID. 8 of them were [group 1, e.g., old adults] and 8 of them were [group 2, e.g., young adults]. After several days, 8 of them got so sick with COVID that they had to go to the hospital. Using the options below, please tell us who the 8 people that got very sick with COVID are." Participants selected from a drop-down menu the number of members from each group who went to the hospital, where the number totaled 8 (e.g., 5 old adults and 3 young adults). The order of comparisons and the order in which we presented the groups in each comparison were randomized.

Finally, participants were asked what conversations they might have with their child regarding race, social class, and age disparities (with race, social class, and age questions in three counterbalanced blocks). Before receiving the questions, participants were asked to focus on their child (5–12 years old) and create a pseudonym (to ensure privacy and that no identifying names were included in the Qualtrics output) that would be embedded in the following questions. If participants had more than one child between 5 and 12 years old, they were asked to focus on just one. Each block began by providing the participant with factual information about the disparity in that block, specifically, the rates at which people in contrasting groups in the US were hospitalized for COVID-19-related reasons. For example, for the age disparity, parents read: "A recent study of COVID cases in the United States found that older adults (75-84 years old) are 8 times more likely to be hospitalized due to COVID-19 than young adults (18-29 years old)." For the race disparity, parents read that Black people were 2 times more likely to be hospitalized due to COVID-19 than White people. For the social class disparity, parents read that people living in the poorest zip codes were 3 times more likely to be hospitalized due to COVID-19 than people living in the richest zip codes. This information was based on statistics provided by the CDC for age and racial differences [2,3] and Owens for social class differences [4].

Next, they received a self-reported **Discussion Likelihood** question: "Would you talk about this difference with [child pseudonym]?" and responded on a 5-point Likert scale (1 = definitely no, 3 = might or might no, 5 = definitely yes). Due to a typographical error, the answer choice appeared as "might or might no" rather than "might or might not." However,

participants appeared to have no difficulty understanding what was being asked. If they selected 3, 4, or 5, they were asked, "What might you say to [child pseudonym]?" Regardless of their first response, participants were asked, "If [child pseudonym] asked you why [older adults, Black people, poor people] are more likely to go to the hospital from COVID-19 than [young adults, White people, rich people], how would you explain it?" and "Why would you talk or not talk to [child pseudonym] about this information?"

After completing all three blocks, participants were asked to complete four survey measures (Stereotype Content, Internal Motivation to Respond without Prejudice Scale, Category Disregard, and Social Constructionist and Essentialist Beliefs Scale; see below), complete a short demographics form, then were debriefed and received compensation for their participation. The study took about 25 minutes, and each participant received $3.80.

**Stereotype content (adapted from [85]).** For each of six social groups (Black people, White people, poor people, rich people, young people, old people), participants indicated how strongly they agreed with each of 6 attributes (3 for warmth, 3 for competence) using a 5-point Likert scale (1 = not at all, 5 = extremely). Sample items for warmth and competence include, "As viewed by society, how sincere are members of this group?" and "As viewed by society, how effective are members of this group?", respectively. We observed good internal consistency for each social group, with all αs ≥ .86 (warmth) and ≥ .85 (competence).

**Internal Motivation to Respond Without Prejudice Scale (IMS) [49].** This measure included 5 items that were slightly modified to assess IMS generally across all social groups, rather than only toward Black people (as originally written). We did not assess IMS separately for age, race, and social class, as doing so would have required assuming which category in each dimension (e.g., old vs. young? rich vs. poor?) participants might be more prejudiced about. Participants indicated how strongly they agreed with 5 statements using a 7-point Likert scale (1 = strongly disagree, 7 = strongly agree). One sample item is, "Being nonprejudiced is important to my self-concept." We observed acceptable internal consistency for the IMS items, α = .75.

**Category Disregard (G/RADS) [57].** The G/RADS measures personal and societal acknowledgment and disregard of categories, but given our interest in parents' individual beliefs, only the personal disregard subset of items from the G/RADS was adopted. For each social category (race, social class, age), participants indicated how strongly they agreed with each of 5 statements on a 7-point Likert scale (1 = strongly disagree, 7 = strongly agree). A sample item includes, "A person's race/social class/age isn't relevant to how I treat them." We observed good internal consistency for all category disregard items for race, social class, and age, with all αs ≥ .86.

**Social Constructionist and Essentialist Beliefs Scale (SCEBS) [68].** This measure included 13 items for race, 13 items for social class, and 11 items for age. For race and social class, participants indicated how strongly they agreed with 13 statements (7 for social constructionist beliefs, 6 for essentialist beliefs) using a 7-point Likert scale (1 = strongly disagree, 7 = strongly agree). For age, participants indicated how strongly they agreed with 11 statements (7 for social constructionist beliefs, 4 for essentialism) using a 7-point Likert scale. Two essentialism items were removed for age because the language was not compatible (e.g., "A person's core essence can't change; it is always affected by their age regardless of what they do"). We excluded the realism subscale of SCEBS because the items do not apply to the current study context (e.g., "Age is a real category"). Sample items for social constructionist and essentialist beliefs include, "How we talk about [race/social class/aging] as a society affects what [race/social class/aging] actually is," and "It is possible to predict many aspects of a person once you know their [race/social class/age]," respectively. We observed acceptable or good internal consistency for all items for race, social class, and age; all αs ≥ .84 (social constructionist) and ≥ .75 (essentialist).

## Qualitative coding

We used a codebook approach to code the content of parents' explanations, the strategies they might use to discuss each disparity, and the reasons they would or would not discuss the disparities with their children. Our coding process

 

started with deductively identifying codes based on established codebooks in the literature, and then examining whether additional inductive codes were necessary to accurately represent the data. For parents' explanations of the relation between social categories and illness, we used a coding scheme developed by Menendez et al. [12], which categorized explanations of health disparities into biological, behavioral, structural, or everyone-equal explanations. The first and second author read all the responses, and to better capture participants' responses, adapted the coding scheme to include a code for explanations and preventive behaviors. See Table 1. For parents' strategies, we used coding schemes from previous studies examining parent-child conversations (specifically, reassurance, deflection, and would not answer) [8] and parental explanations (specifically, search for information and expressing uncertainty) [86]. When reading through the responses, the first and second authors noticed that many parents mentioned not believing the information provided or even rejecting it. Therefore, we included a code to capture those responses as well. See Table 2. For parents' reasons for discussing health disparities, the first and second author read all the responses and noticed that parents tended to mention three main reasons: importance, prompted, and shielding. See Table 3.

**Table 1. Qualitative coding of explanations.**

| Code | Description | Example | % Agreement | κ |
|---|---|---|---|---|
| **Explanation** | Answer provides an explanation for the difference. | "Because they [poor people] don't have much money for medication" | 90.4 | .82 |
| **Biological** | Mentions the body, immune response, pre-existing condition, genetics, comorbidities, healthiness/unhealthiness. | "They have weaker immune system when compared to younger people" | 97.3 | .94 |
| **Behavioral** | Mentions actions and behaviors individuals or groups partake in (past, present, and future). This does not include if they say that the child should engage in a behavior. | "Because they [poor people] don't protect themselves and stay 6 feet away from people at all times." | 98.1 | .70 |
| **Structural** | Mentions structural factors that may lead to individuals contracting illnesses. Structural factors include, but are not limited to, racism, poverty, working conditions, living conditions, access to healthcare, access to food, or the environment. | "Poor people don't have as much access to healthcare and living in poverty might restrict some from seeking a vaccine" | 95.1 | .86 |
| **Everyone is equal** | Mentions that both groups are humans, both are people, both are equal, viruses do not discriminate between people or different social categories (e.g., race), or anyone can get sick. | "I would say it doesn't matter white or black it's what their lifestyle looks like" | 96.6 | .93 |
| **Preventive** | Mentions that the child or the family should engage in preventive behaviors to protect themselves or other members of the family (e.g., grandparents). | "That the areas outside where we live may be more affected and to stay away" | 95.1 | .71 |

*Note:* The kappa reported for the "Everyone is equal" code was calculated using PABAK.

**Table 2. Qualitative coding of strategies.**

| Code | Description | Example | % Agreement | κ |
|---|---|---|---|---|
| **Reassurance** | Attempt to comfort the child or help manage their emotions | "That we will be ok and things will be ok" | 97.0 | .63 |
| **Deflection** | Shift the conversation to a different direction | "Talk about it later" | 95.6 | .61 |
| **Would not answer** | Parents says that they would not answer | "I wouldn't because I wouldn't even know how to start or what to talk about" | 98.1 | .89 |
| **Rejection of information** | States the information is invalid, not true, or that it is racist or fake news. | "I don't believe these facts are true. Colour has nothing to do with anything" | 98.9 | .89 |
| **Search for information** | Parent says that they will Google, search the study, or look up more information | "I would look up the research with him so we both know why." | 99.6 | .95 |
| **Expressing uncertainty** | Parents says that they don't know enough, can't explain it, don't know the reason why, or just respond with I don't know | "I really don't know how to explain it" | 97.0 | .93 |

**Table 3. Qualitative coding of reasons.**

| Code | Description | Example | % Agreement | κ |
|------|-------------|---------|-------------|---|
| **Important** | Parent says they want to talk about it with their children because the information is important for the children to understand the world, personally relevant because of who they are, or relevant to the family | "Because it is important for him to know and I wouldn't want him to get covid" | 87.0 | .62 |
| **Prompted** | Parents state that they would only talk about the topic if the child asks or starts the conversation | "If he asks then I'll talk about it" | 97.4 | .83 |
| **Shielding** | Parent says that they want to protect the child from information because: they will not understand, they are not old enough, they don't want to upset them or scare them, they do not need to know/not relevant, important, necessary, or not an issue for the child. | "Some information I would not talk about because he is not old enough to understand" | 96.7 | .89 |

Because parents received three open-ended questions per disparity, codes were assigned when they were present in any of their responses for each disparity. Each set of parent responses could receive multiple codes, as applicable. Two trained coders coded 20% of the responses (chosen randomly) to determine the reliability of each code. One code (everyone is equal) was not deemed reliable. Upon inspection, this seemed to be a prevalence issue, as the percent agreement was high. For this code, we calculated a prevalence-adjusted and bias-adjusted kappa (PABAK) [87], which showed that after adjusting for prevalence, the coding showed substantial agreement. All other codes showed moderate to substantial agreement (kappa > .60), and disagreements were resolved through discussion (see Tables 1–3 for descriptions, examples, and reliability statistics).

## Results

The results are organized into four sections, corresponding to the four research questions listed in the Introduction. Our data file, analysis scripts, and Supporting Material can be found at: https://osf.io/jv9at/?view_only=6a98e72193d64fb2b037f118793c5e55

### RQ1: Discussion likelihood

In this section, we examined: (a) What are parents' likelihood of saying they would discuss COVID-19 health disparities with their children? (b) Does their likelihood depend on their child's age? (c) Does their likelihood depend on whether the disparity is about age, race, or social class?

To examine these three questions, we preregistered fitting an ordinal mixed-effects regression predicting parents' likelihood of discussing a COVID-19 health disparity (on a 1–5 scale), with disparity type (age, race, social class) and their child's age as fixed effects, and participant as a random effect. When an effect of disparity type was present, we used dummy code contrasts to determine which disparities parents were more likely to discuss. Parents' self-reported likelihood of discussing the health disparities differed based on the disparity type ($\chi^2_{LR}(2) = 48.84$, $p < .001$) and their child's age ($\chi^2_{LR}(1) = 171.77$, $p < .001$). Parents were significantly more likely to report that they would discuss the age disparity ($M = 3.64$, $SD = 1.28$) than either the race (OR = 2.51, $M = 3.20$, $SD = 1.33$, $p < .001$) or social class disparity (OR = 1.97, $M = 3.31$, $SD = 1.20$, $p < .001$). There was no significant difference between the likelihood of parents discussing the race and social class disparities (OR = 1.28, $p = .057$). Parents of older children were significantly more likely to report that they would discuss these disparities than parents of younger children (OR = 1.16, $p = .002$). The $\chi^2_{LR}$ were obtained in R [88] using the *RVAideMemoire* package [89].

We also conducted three non-preregistered analyses to further interrogate these results. First, we explored parents' likelihood of discussing the health disparities as a function of demographic variables, including parental political affiliation, political leaning, and education level, in three separate regression models, one per disparity. Parents who indicated 'no political affiliation' or 'other' were excluded from this analysis due to their small group size ($n = 17$). For the age health

disparity, Democrats (*M* = 3.88) were significantly more likely to say they would discuss the disparity than Independents (OR = 1.81, *M* = 3.54, 95% CI [1.18, 3.37], Wald *z*-test = 2.70, *p* = .02) and Republicans (OR = 1.96, *M* = 3.37, 95% CI [1.14, 3.40], Wald *z*-test = 2.88, *p* = .01). For the race health disparity, Democrats (*M* = 3.45) were significantly more likely to say they would discuss the disparity than Republicans (OR = 1.87, *M* = 2.95, 95%CI [1.09, 3.21], Wald *z*-test = 2.72, *p* = .02) but not Independents (OR = 1.58, *M* = 3.08, 95% CI [0.95, 2.63], Wald *z*-test = 2.09, *p* = .09). For the social class health disparity, Democrats (*M* = 3.60) were significantly more likely to say they would discuss the disparity than Republicans (OR = 2.54, *M* = 2.93, 95% CI [1.44, 4.48], Wald *z*-test = 3.87, *p* < .001), but not Independents (OR = 1.65, *M* = 3.24, 95% CI [0.99, 2.75], Wald *z*-test = 2.31, *p* = .05. Furthermore, when controlling for the individual difference measures previously discussed, the difference between Democrats and Republicans remained significant for age (OR = 2.02, 95% CI [1.12, 3.63], Wald *z*-test = 2.83, *p* = .01) and social class (OR = 2.21, 95% CI [1.22, 4.02], Wald *z*-test = 3.13, *p* = .004), but not race (OR = 1.49, 95% CI [1.19, 2.66], Wald *z*-test = 1.65, *p* = .22). No other effects were significant. Wald tests were obtained using the *ordinal* package [90].

Second, we conducted non-preregistered two-sample *t*-tests (one per comparison) to assess whether parents' disparity judgments (forced-choice) predicted their likelihood of discussing each disparity. For each, we compared parents who thought there was a disparity (by selecting either of the two social groups as more likely to get very sick with COVID) with those who reported there was no difference. Parents who thought there was a social class disparity were significantly more likely to say they would discuss the disparity (*M* = 3.56) than parents who did not believe there was such a disparity (*M* = 3.32, *p* = .026). There were no significant differences in discussion likelihood for the age (*p* = .174) and race (*p* = .054) disparities, as a function of their disparity judgment (forced-choice).

Third, we fit three non-preregistered ordinal logistic regressions (one per comparison) to determine whether endorsing any of the three explanations for each disparity predicted parents' likelihood of discussing that disparity with their children (see Table 4). For the age disparity, parents who endorsed the biological explanation were significantly more likely to say they would discuss the disparity (OR = 2.43, 95% CI [1.60, 3.72], Wald *z*-test = 4.151, *p* < .001), but there were no differences for endorsing the behavioral (OR = 1.19, 95% CI [0.78, 1.79], Wald *z*-test = 0.866, *p* = .387) or the structural explanation (OR = 1.28, 95% CI [0.83, 1.98], Wald *z*-test = 1.14, *p* = .256). For the race disparity, parents who endorsed the structural explanation were significantly more likely to say they would discuss the disparity (OR= 1.51, 95% CI [1.01, 2.26], Wald *z*-test = 2.01, *p* = .045), but there were no differences for endorsing the behavioral (OR = 1.42, 95% CI [0.94, 2.16], Wald *z*-test = 1.69, *p* = .09) or the biological explanation (OR = 1.16, 95% CI [0.81, 1.67], Wald *z*-test = 0.82, *p* = .41). Finally, for the social class disparity, parents who endorsed the behavioral explanation were significantly more likely to say they would discuss the disparity (OR = 1.77, 95% CI [1.19, 2.65], Wald *z*-test = 2.80, *p* = .005), as did those who endorsed the structural explanation (OR = 1.49, 95% CI [1.02, 2.19], Wald *z*-test = 2.06, *p* = .039), but there were no differences for endorsing the biological explanation (OR = 1.16, 95% CI [0.81, 1.65], Wald *z*-test = 0.84, *p* = .40).

**Table 4. Parents' mean discussion likelihood scores as a function of their explanation endorsements (yes or no).**

| Disparity | Behavioral | | Biological | | Structural | |
|---|---|---|---|---|---|---|
| | Yes | No | Yes | No | Yes | No |
| Age (*n*) | 3.80 (147) | 3.56 (291) | *3.76 (357)* | *3.12 (85)* | 3.84 (110) | 3.57 (332) |
| Race (*n*) | 3.49 (140) | 3.06 (301) | 3.29 (279) | 3.05 (163) | *3.51 (139)* | *3.06 (301)* |
| Class (*n*) | *3.62 (178)* | *3.11 (262)* | 3.40 (265) | 3.18 (175) | *3.58 (183)* | *3.13 (256)* |

*Note:* Cells showing significant differences as a function of explanation endorsement are indicated in bold italics.

**RQ2: Disparity judgments (forced-choice and scale)**

In this section, we examined two questions: (a) Do parents think social categories influence the likelihood of someone getting very sick with COVID-19? (b) How would parents explain these disparities?

The disparity judgment (forced-choice) task assessed parental rates of endorsing disparities based on age, race, and class. For each comparison type, we preregistered comparing the percentage of parents who gave each response against chance (33.33%) using a *t*-test. We used a Bonferroni correction to adjust for multiple tests, using an alpha level of .017 (.05/3). Results showed that parents' choices all differed significantly from chance (33.33%), although some were significantly above, and some were significantly below. Specifically, choices were significantly above chance levels when choosing "old adults" for the age comparison and "they are the same" for the other comparisons, and choices were significantly below chance levels for all other responses (see Table 5).

For the disparity judgment (scale) task, we preregistered fitting a linear regression (one per comparison) predicting parents' confidence ratings (1−4) when they reported one group was more likely to get sick or both were the same, with "they are the same" as the reference group. For the age comparison, there was no difference in confidence between parents who chose old or young adults (*M* = 3.14) or "they are the same" (*M* = 3.12) (*F*(1,441) = .07, b = 0.02, *p* = .80). For the racial comparison, parents who chose Black or White people (*M* = 3.10) were significantly less confident than parents who chose "they are the same" (*M* = 3.38) (*F*(1,441) = 7.38, b = −0.28, *p* = .007). For the social class comparison, parents who chose poor or rich people (*M* = 3.08) were significantly less confident than parents who chose "they are the same" (*M* = 3.30) (*F*(1, 441) = 5.71, b = −0.22, *p* = .0173). For the personality comparison, parents who chose nice or mean people (*M* = 2.96) were significantly less confident than parents who chose "they are the same" (*M* = 3.35) (*F*(1, 441) = 12.98, b = −0.55, *p* < .001). T-tests and linear regression results were obtained using the *stats* package [88].

We preregistered calculating how many members of the objectively more likely groups (i.e., old adults, Black people, and poor people) parents believed would go to the hospital and tested whether these scores were significantly different

**Table 5. Parents' choices of which group is likelier to contract COVID-19.**

| Comparison Type | % of participants | t(442) |
|---|---|---|
| **Old or young adults** | | |
| Old adults | 72.91*** | 18.88 |
| They are the same | 24.15 *** | −4.35 |
| Young adults | 2.94 *** | −37.45 |
| **Black or White people** | | |
| Black people | 11.29 *** | −14.43 |
| They are the same | 85.55 *** | 31.43 |
| White people | 3.16 *** | −35.86 |
| **Poor or rich people** | | |
| Poor people | 22.35 *** | −5.38 |
| They are the same | 76.3 *** | 21.41 |
| Rich people | 1.35 *** | −57.56 |
| **Nice or mean people** | | |
| Nice people | 1.81 *** | −49.25 |
| They are the same | 94.13 *** | 54.68 |
| Mean people | 4.06 *** | −30.81 |

*Note:* Parents' choice rates for who was more likely to get sick with COVID-19, when given a choice between two social groups or "they are the same", per comparison.

All comparisons to chance (33.33%): *** *p* < .001

from the scale midpoint (4.0), using *t*-tests. Although there is no objectively more likely group between mean and nice people, we used mean people as the comparison case for analysis purposes, as they were the group selected more in the forced-choice task. Results showed that parents selected old adults, poor people, and mean people at levels greater than chance (4.0). Selections of Black vs. White people were not significantly different from chance (see Table 6).

We also conducted a non-preregistered comparison of how often participants were consistent across the two disparity judgment tasks (forced-choice and scale). For each participant, we tallied whether or not their forced-choice response on a given comparison type matched their response on the 8-point Likert scale. To count as a match, a choice of one category needed to correspond to selecting at least 5 members of that category on the scale, and a choice of both categories needed to correspond to selecting exactly 4 members of that category on the scale. On this metric, parents' responses matched on 70% of trials overall (age: 60%; race: 75%; social class: 70%; personality: 75%). Most discrepancies occurred when parents who reported that there was a difference on the forced-choice task (e.g., reporting that old adults were more likely to get sick than young adults) selected exactly 4 members out of 8 on the scale task. This may indicate that, when they reported a disparity on the forced-choice task, it was thought to be a small one (less than 62.5% [5/8] vs. 37.5% [3/8], which is the minimum that could be detected on the scale task). It is also possible that the difference reflects how the two tasks were framed, as the forced-choice task was about getting "very, very sick" with COVID, whereas the scale task was about hospitalizations from COVID. See Figure S1 in S1 File.

Table 7 shows the proportion of participants who endorsed each explanation for each of the three documented disparities (age, race, and class). We preregistered fitting mixed-effects logistic regressions predicting whether parents indicated "yes" to a given explanation, with explanation type (behavioral, biological, structural) as a fixed effect, and participant as a random effect. We conducted two regression analyses per comparison: one analyzing the endorsements of parents who selected one of the two groups as more likely to get sick, and the second analyzing the endorsements of parents who selected "they are the same" (see Tables S1 and S2 in S1 File). Overall, these analyses show that parents explained age and race disparities more often by appealing to biological factors, and race and class disparities by appealing to structural factors. [2] were obtained using the *lme4* package [91] and differences in endorsement by explanation type were obtained using the *eemeans* package [92].

## RQ3: Parents' explanations

Qualitative coding analysis revealed that some codes appeared more commonly for some disparities than others. To determine if these were stable differences, we conducted a non-preregistered mixed-effects logistic regression separately for each qualitative code, examining whether the presence of each code differed by disparity type as a fixed effect and participant as a random effect. We conducted pairwise comparisons to determine the differences between disparity types. There are several notable patterns regarding how frequently parents' responses include each code (see Table 8). First,

**Table 6. Disparity judgment (scale) task, mean number of selections of the bolded group, out of 8 possible.**

| Comparison Type | *M* (*SD*) | *t*(441) |
|---|---|---|
| **Old adults** or young adults | 5.02 (2.08) *** | 10.27 |
| **Black people** or White people | 4.08 (1.43) | 1.19 |
| **Poor people** or rich people | 4.33 (1.63) *** | 4.19 |
| **Mean people** or nice people | 4.14 (1.45) * | 1.99 |

*Note:* Comparisons to scale midpoint (4.0):

* *p* < .05,

**\*\*p* < .01,

\*\*\*p* < .001

**Table 7. Proportion of explanations endorsed per comparison type, separately by response on the disparity judgment (forced-choice) task.**

| Comparison Type/ Choice | n | Explanation Type | | |
|---|---|---|---|---|
| | | Behavioral | Biological | Structural |
| **Age** | | | | |
| Old or young adults | 335 | .30a | **.88b** | .22c |
| They are the same | 107 | .42a | **.59b** | .36a |
| **Race** | | | | |
| Black or White people | 64 | .36a | **.59b** | **.64b** |
| They are the same | 376-378 | .31a | **.64b** | .26a |
| **Social Class** | | | | |
| Poor or rich people | 105 | .56a | .49a | **.91c** |
| They are the same | 334-335 | .36a | **.64b** | .26c |

*Notes*: Within each line, different letters following the proportions indicate significant differences as a function of explanation type. Explanations that were significantly most frequently endorsed appear in bold. P-values were adjusted using Tukey's method for comparing a family of 3 estimates within each comparison of explanation endorsement per disparity judgment. For all significant differences, $ps < .05$. Some comparison type/choices have a range of ns, given different numbers of responses for different reasons (behavioral, biological, and structural).

**Table 8. Proportion of parent responses including each code.**

| | | Age | Race | Social Class |
|---|---|---|---|---|
| Content | | | | |
| | Explanation | **.77a** | .40b | .65c |
| | Biological | **.71a** | .12b | .03c |
| | Behavioral | <.01a | **.06b** | **.06b** |
| | Structural | <.01a | .13b | **.50c** |
| | Everyone is equal | .02a | **.10b** | .06c |
| | Preventive | **.18a** | .12b | .12b |
| Strategies | | | | |
| | Reassurance | <.01a | **.01a** | <.01a |
| | Deflection | 0a | **.02b** | <.01ab |
| | Would not answer | .05a | **.13b** | .09c |
| | Rejection of information | <.01a | **.12b** | .05c |
| | Search for information | 0a | **.03b** | <.01a |
| | Expressing uncertainty | .08a | **.28b** | .14c |
| Reasons | | | | |
| | Important | **.21a** | .13b | .16ab |
| | Prompted only | .05a | .05a | **.06a** |
| | Shielding | .10a | .14ab | **.15b** |

*Note*: Within each line, different letters following the proportions indicate significant differences as a function of code type. Codes that were significantly most frequently endorsed appear in bold. P-values were adjusted using Tukey's method for comparing a family of 3 estimates within each comparison of code per disparity judgment. For all significant differences, $ps < .05$.

parents most often explained the age disparity and least often explained the race disparity. These results align with our finding that parents were most likely to say they would discuss the age disparity compared to the race and social class disparity. Second, most responses for the age disparity included the biological code, whereas the structural code was most common for the social class disparity. This pattern mirrors the proportion of parents who endorsed biological and

structural reasons for the respective disparity in the Explanation Endorsement Task. Finally, the types of messages that indicated some difficulty in engaging with the topic (i.e., rejection of information, expressing uncertainty) or completely evading discussion (i.e., would not answer, deflection) were most common for the race disparity.

### RQ4: Discussion likelihood by individual differences

We preregistered fitting an ordinal logistic regression (one per comparison) predicting the likelihood of parents saying they would discuss each disparity with their children, with the individual difference measures assessing parental beliefs and attitudes toward the target groups, described earlier, as predictors (see Table 9 for odds ratios and significance tests and Tables S33-S35 for descriptives and bivariate correlations in Supporting Material). For these analyses, child age was also preregistered as a predictor in the models. Wald *z*-tests were obtained using the *ordinal* package [90].

## Discussion

Given well-documented health disparities in the US, an important open question is whether and how parents discuss these disparities with their elementary-school-aged children, who are actively constructing their beliefs about social categories as well as contagious illness during this period of development. The COVID-19 pandemic created a unique and especially urgent environment, as parents became their children's most immediate source of information during lockdowns and school closures. To investigate these issues, the current study examined parents' self-reported likelihood of discussing COVID-19 disparities with their children, how parents said they would explain such differences, and how their responses corresponded to their own beliefs and attitudes toward these social groups. We focused on three disparities in particular: age (old vs. young adults), race (Black vs. White), and social class (rich vs. poor), and surveyed parents of children ages 5–12. The current study replicates previous findings on adults' and parents' perceptions of COVID-19 disparities and contributes novel findings to the parental socialization literature. Namely, parents' likelihood of discussing disparities differed by disparity type and their child's age, parents' knowledge and awareness of the disparities, and parents' beliefs and attitudes about social categories and social groups. Finally, the type of explanations that parents reported they would provide to their children differed by the disparity in question. Below, we discuss these findings and their implications for the psychological literature, future interventions, and public policy more broadly.

**Table 9. Odds ratios and significance tests for individual difference measures on the likelihood of discussing each health disparity.**

| Measure | Age | | | Race | | | Social Class | | |
|---|---|---|---|---|---|---|---|---|---|
| | OR | z | p | OR | z | p | OR | z | p |
| **Essentialism** | 1.14 | 1.25 | .21 | 1.34 | 3.24 | .001 | 1.54 | 4.35 | <.001 |
| **Social constructionism** | 1.13 | 1.01 | .31 | 1.24 | 2.01 | .044 | 1.16 | 1.21 | .22 |
| **Category Blindness** | 0.91 | −1.20 | .23 | 0.91 | −0.94 | .35 | 0.90 | −1.34 | .18 |
| **IMS** | 1.17 | 1.70 | .09 | 1.12 | 1.06 | .29 | 1.11 | 0.96 | .34 |
| **Child age** | 1.11 | 2.77 | .006 | 1.09 | 2.21 | .03 | 1.07 | 1.72 | .09 |
| | Old/ Young | | | Black/ White | | | Poor/ Rich | | |
| **Warmth** | 0.89/ 0.80 | −0.91/ −1.42 | .36/ .15 | 1.65/ 0.98 | 2.77/ −0.14 | .005/.89 | 0.96/ 1.12 | −0.25/ 1.07 | .80/ .29 |
| **Competence** | 1.16/ 1.31 | 1.18/ 1.81 | .23/ .07 | 0.64/ 0.91 | −2.55/ −0.58 | .01/ .56 | 0.85/ 0.98 | −1.03/ −0.22 | .30/ .83 |

**Note:** Social constructionism, essentialism, category blindness, and IMS items were scored on a 7-point scale. Warmth and Competence items were scored on a 5-point scale. IMS assessed motivation to respond without prejudice generally across all social categories.

## Parental likelihood of discussing disparities with their children

In our data, we found that parents were moderately likely to say they would discuss COVID-19 health disparities related to age, race, or class with their children (with means between 3 and 4, on a 1–5 scale), but responses differed based on their child's age and the disparity type. Parents of older children reported they were more likely to discuss any of the three disparities than parents of younger children. Some parents expressed the belief that younger children were not socially, cognitively, or emotionally prepared to discuss these issues (e.g., "At this age, she doesn't need to know about the class system we have in place. I feel that this is not a topic that should be discussed with a 7-year-old"; "I would not because she has grandparents this age and I would not want to freak her out about losing her grandparents to Covid"). This is consistent with prior work showing that parents tend to shield younger children from information about COVID-19 [8].

Parents were significantly more likely to say they would discuss the age disparity than the race or social class disparity (with no differences between the latter two). This pattern may reflect that, prior to learning about these disparities in our survey, parents were substantially more aware of age-based disparities (acknowledged by 77% of the sample) than either race- or class-based disparities (acknowledged by only 12% and 22% of the sample, respectively), consistent with prior research [10–12]. Partially, this could be due to the fact that the disparities are indeed much greater for age than for race or social class (as also reflected in the factual information provided to participants). However, it is notable that, even after receiving the statistics about the disparities, some parents rejected this information (e.g., "I feel it's a made up statistic"; 12% for race and 5% for class, as compared to <1% for age), perhaps due to ancillary beliefs about race and class that conflicted with the disparity facts. Parents' reluctance to discuss disparities associated with race (and to some extent class) may also have been due to uncertainty about what they should say (e.g., "I would tell him im (sic) honestly not sure myself": 28% for race, 14% for class, and 8% for age). These findings are consistent with prior research suggesting that some White parents do not feel responsible for discussing topics about race, even when racial differences are made salient, and avoid in-depth conversations about race [16,22,23,58]. Some parents may also believe that younger children may not fully understand racial topics [24,93] -- despite ample scientific evidence that children as young as four years of age are aware of social inequalities and seek to rectify them [36,37,39–41].

Another potentially relevant factor responsible for higher likelihoods of talking about age is that the adverse health outcomes associated with older adults could in theory affect any family, including close family members of this primarily White, relatively well-educated sample. In their open-ended responses regarding age, some parents did make reference to older family members (e.g., "We need to be careful around older adults like grandpa. He would get much sicker with covid than us"; "I would talk to her regarding this because she has grandparents and a great grandma"). In contrast, for race and class, the adverse outcomes are specifically associated with minority groups [94] and thus may not have been viewed as personally relevant to some participants.

Our findings also provide initial insights into parents' thoughts about social class disparities–a relatively neglected topic in the literature. More research is needed to better understand why parents are less aware of, and less likely to discuss the class disparity than the age disparity, in the broader context of parents' messages to their children regarding social class.

## Parental beliefs about the disparities

Our finding that parents were more sensitive to age disparities than race or social class disparities replicates prior work [10,12]. It also extends the literature by showing that parents who believed race and social class do not influence the likelihood of getting seriously sick with COVID were in fact more confident in their judgments than those who believed that a disparity existed. No such difference was found for the questions regarding age, which might be due to parents' stronger belief that age disparities existed. Parents' explanation endorsements provide some insight into why they reported that there were no disparities based on race or social class. Most typically, they said that the lack of such differences was for biological reasons, namely, that all humans are biologically alike (e.g., "I would tell her that everyone is equal").

In contrast, those who did report group differences based on race or class were substantially more likely to endorse structural explanations. At the same time, it is important to note that those parents who acknowledged a disparity based on race or social class also endorsed biological explanations relatively often (about half the time), consistent with prior research indicating that parents rely on multiple explanatory frameworks for group differences [12].

## How parental beliefs about the disparities related to their likelihood of discussing disparities with their children

As assessed by the disparity judgment (forced-choice) task, parents' beliefs about the causes of health disparities were related to their self-reported likelihood of discussing these disparities with their children. For the age disparity, parents who endorsed *biological* explanations (compared to those who did not) were more likely to say they would discuss this disparity with their children. For race and class disparities, parents who endorsed *structural* explanations (compared to those who did not) were more likely to say they would discuss these disparities with their children. This suggests that parents who are aware of and knowledgeable about external factors contributing to adverse health outcomes in Black and low-income individuals, such as limited access to quality health resources [95,96], may be more comfortable engaging in conversations about differences among social groups with their children. Additionally, for the class disparity, parents who endorsed *behavioral* explanations (compared to those who did not) were more likely to say they would discuss this disparity with their children. It is possible that parents thought that structural differences would lead to behavioral differences which could then relate to health outcomes. However, it is also possible that parents who believed class disparities were due to behavior reasoned that explaining these behavioral differences to their children would be easy, leading them to be more likely to discuss this health inequality. Future work should examine how parents' causal reasoning about health disparities relates to specific motivations for discussing inequalities.

## How parental individual differences related to their likelihood of discussing disparities with their children

For each of the disparities (age, race, and social class), we also examined how parental discussion with their children related to their essentialist beliefs, social constructionist beliefs, stereotype content, internal motivation to respond without prejudice, and category disregard. The findings were complex and differed by disparity. For the age disparity, none of these factors predicted parental likelihood of discussing health disparities involving age (which again can indicate that across different demographics, parents were generally comfortable discussing age disparities). For the social class disparity, the only predictor was essentialism. It may be that parents with more essentialist views about social class attributed the disparities to internal characteristics and thus viewed them as relatively straightforward to explain, even if incorrect. This would be consistent with prior research on social class beliefs in the US, in which being poor is often viewed as related to character rather than structural issues [97]. Future work should investigate this possibility.

In contrast, for the race disparities, several factors were predictive of parents' likelihood of discussing this disparity with their children: social constructionism, essentialism, and stereotype content (specifically, viewing Black people as warmer, and as less competent). This constellation of factors is surprising on the face of it and difficult to explain, given that social constructionism and essentialism are typically assumed to be negatively correlated with one another [68], and because the ratings for warmth and competence yielded opposite effects. Perhaps these findings reflect that different parents have different motivations for discussing (or not discussing) race disparities, with some parents (i.e., those with higher social constructionist beliefs and more positive views of Black warmth) doing so because they are sensitive to the role of structural forces that contribute to race disparities, and other parents (those who score *lower* on essentialism, and who are *less* likely to endorse stereotypes about the lower competence of Black people) *avoiding* such discussion because they have a 'race-neutral' ideology according to which such group differences are not appropriate to discuss. However, this explanation remains very speculative and will require further research to determine if these findings replicate and, if so, to better understand the reasons for these patterns.

We also examined how the likelihood of discussing the disparities with their children related to parents' demographics (education, political affiliation, and political leaning). For all three disparities (age, race, and social class), we found that political affiliation was predictive: Democrats were significantly more likely to discuss these disparities with their children than were Republicans. These differences were still present after controlling for individual difference factors for the age and social class health disparities, but not the race health disparity. These findings align with prior work showing that Democrats tend to acknowledge COVID-19 health disparities related to age, race, and social class more often than Republicans [10,11,98]. Therefore, it is plausible that Republican parents are less likely to have these conversations with their children if they do not believe the disparities exist. The finding that parents' likelihood of discussing the race disparity did not differ by political affiliation when controlling for individual differences suggests that discussions about racial issues may be related more to parents' beliefs about race [53–55] or their racial socialization practices [12], but more work is necessary to clarify this pattern.

## Implications

Our findings have implications for informing future research, intervention, and public policy. Most notably, it is striking that parents generally reported that they did not believe there is a difference in people's likelihood of contracting COVID-19 based on race or social class, and that they did not fully consider how structural factors contribute to such disparities. Certainly, if parents do not believe a disparity exists, they will not talk with their children about it, and if they do not explain the role of structural factors, children may attribute such differences to behavioral choices or even essentialist underpinnings, both of which could result in negative stereotyping. A future direction that we think would be highly informative is to provide parents with factual information about social determinants of health, to determine whether such information affects their own understanding, as well as how they discuss these issues with their children. Additionally, our qualitative results suggest obstacles to parental discussion of the topic, with some expressing uncertainty about how to engage their children, or a belief that children are developmentally unprepared for conversations about race and social class. Future interventions should also consider how best to communicate to parents their children's early attention to different social groups within their culture, as well as the value of providing them with accurate explanatory frameworks for differences associated with race [93] and social class [40]. Additionally, given the politicization of COVID-19 policies and health recommendations, it would also be informative to see how parents' perceptions of COVID-19-related disparities compare to those of other health disparities (both contagious, such as the flu, and non-contagious, such as diabetes). Exploring these questions could help future researchers and public health experts determine how to best engage with parents regarding the structural causes of health disparities.

Our work can also help inform multisectoral policy efforts aimed at communicating the nature of COVID-19 disparities to the general public. According to our and other research [10–12], adults are, in general, not fully aware of the extent of the racial and social class COVID-19 disparities in the US. Therefore, a clear opportunity is available for educators, healthcare experts, public health services departments, and policymakers to collaborate to deliver comprehensive, evidence-based programming on illness disparities in the US. For example, the extent to which healthcare workers are aware of the impact of non-biological and non-behavioral factors on an individual's contagion risk is unknown and could be a source for informing adults. Additionally, efforts in educational spaces (e.g., schools, museums) could contribute to children's developing beliefs about health disparities as they spend more time away from their parents. For example, educators and health professionals could collaborate to teach children about the various health and non-health related factors that can affect someone's likelihood of contracting an illness. This leaves an open area for educators and public health experts to create educational programming to provide the most up-to-date research on social determinants of health.

## Limitations

The current study offers valuable insights into parents' beliefs about COVID-19 health disparities and how they think they would discuss these important topics with their children. Despite the many findings, the current study had several limitations and unanswered questions. First, parents only provided self-reports of what they thought they would say to

their children, and we did not examine how they actually talked with their children. Thus, an important future question is the accuracy of these self-reports, and how parent-child conversations unfold when the child is actively participating (e.g., responding to the parent, asking questions, offering their own beliefs). Second, social desirability factors may have influenced what parents were willing to report, especially on some of the individual difference measures. Third, the survey itself was long and included multiple parts, so participants' responses to earlier portions of the survey may have influenced their responses to the questions analyzed here or experienced fatigue. For example, prior to being provided with factual information regarding disparities and whether and how they would discuss them with their child, participants first were asked who is more likely to get sick with COVID-19, how confident they were with their choice, and what reasons they would endorse to explain their answer. Additionally, despite the study being open to any parent living in the US, the sample was predominantly White, and over half had earned at least a 2-year college degree. Future research should certainly diversify the sample by race, age, and social class. It would also be important to examine the beliefs and conversations of those individuals who were most affected by COVID-19. Finally, we cannot determine whether parents' likelihood of discussing illness disparities differs by illness type, although ethnic-racial disparities in the severity of other illnesses exist [99]. However, it is noteworthy that when comparing how adults and children reason about COVID-19 and the common cold, very few differences emerged across illness type [12]. More work is required to understand how COVID-19 reasoning differs from other illnesses (e.g., the flu).

## Conclusion

The COVID-19 pandemic offered a unique opportunity to explore whether and how parents would discuss disparities in adverse health outcomes among different groups with their children. Our research concludes that whether and how parents report they would discuss COVID-19 health disparities reflects at least three key factors: (a) their own beliefs about if and why the disparities exist, (b) their attitudes toward the social groups involved, and (c) for many, their discomfort discussing these issues with their children. Overall, these findings suggest that parents may play a role in shaping their children's beliefs about social disparities in the US, informing researchers on where parents stand on discussing disparities and potentially guiding effective methods for explaining these differences.

## Supporting information

**S1 File.  Supporting Material.**
(DOCX)

## Acknowledgments

We would like to thank Nicolena Antifonario-Capello and Linda Treviño in the Cognitive Development Lab for their assistance with qualitative coding. Portions of this project were presented at the Society for the Study of Human Development 2023 conference and the Society for Research in Child Development 2025 conference.

## Author contributions

**Conceptualization:** Lester A. Mejia Gomez, David Menendez, Susan A. Gelman.

**Data curation:** Lester A. Mejia Gomez, David Menendez, Susan A. Gelman.

**Formal analysis:** Lester A. Mejia Gomez, David Menendez, Valerie Umscheid, Susan A. Gelman.

**Funding acquisition:** Susan A. Gelman.

**Investigation:** Lester A. Mejia Gomez, David Menendez, Susan A. Gelman.

**Methodology:** Lester A. Mejia Gomez, David Menendez, Susan A. Gelman.

**Project administration:** Lester A. Mejia Gomez, David Menendez.

**Resources:** David Menendez, Susan A. Gelman.

**Supervision:** Lester A. Mejia Gomez, David Menendez, Susan A. Gelman.

**Validation:** Lester A. Mejia Gomez, David Menendez, Susan A. Gelman.

**Writing – original draft:** Lester A. Mejia Gomez.

**Writing – review & editing:** Lester A. Mejia Gomez, David Menendez, Susan A. Gelman.

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
