## [Decision Letter · Decision Letter 0]

2 Mar 2025

PONE-D-24-52344Who gets sicker and why? Parents’ perceptions of COVID-19 disparities and how they would explain them to their childrenPLOS ONE?

Dear Dr. Menendez,

Thank you for submitting your manuscript to PLOS ONE. After careful consideration, we feel that it has merit but does not fully meet PLOS ONE’s publication criteria as it currently stands. Therefore, we invite you to submit a revised version of the manuscript that addresses the points raised during the review process.

When revising your manuscript, please consider all issues mentioned in the reviewers' comments carefully. Please outline every change made in response to their comments and provide detailed information for any comments not addressed.

Please note that your revised submission may need to be re-reviewed

We look forward to receiving your revised manuscript.

Kind regards,

Emine Ozturk, Ph.D.

Academic Editor

PLOS ONE

2. Thank you for stating in your Funding Statement:  [This research was supported by NSF Grant #2055164 to SAG. Any opinions, findings, and conclusions or recommendations expressed in this material are those of the authors and do not necessarily reflect the views of the National Science Foundation.].Please provide an amended statement that declares *all* the funding or sources of support (whether external or internal to your organization) received during this study, as detailed online in our guide for authors at http://journals.plos.org/plosone/s/submit-now.  Please also include the statement “There was no additional external funding received for this study.” in your updated Funding Statement. Please include your amended Funding Statement within your cover letter. We will change the online submission form on your behalf.

Additional Editor Comments:

No information about the methods was provided in the abstract. Please provide information on the types of quantitative and qualitative data analysis used.Please add information about the demographics of the participants to the abstract.What are your findings of each analysis? Please provide clear and concise information in the abstract.Please avoid any promotional statements in the manuscript. (e.g., "No research has looked at.."). It is not sufficient to indicate that a research topic has not been examined before in a way that the authors will examine it.The current study section should briefly describe the research gap, aims, hypotheses/ research questions (RQs).The rationales should be provided in the literature review, not in the current study section. Avoid any new information or new citations in this section.What are your directional hypotheses?  Please explain them in the "current study" section.Which analysis was used for each hypothesis/ RQs? Justification is needed for each analysis (e.g., mixed effect logistic regression). Please see reviewers comments.The current version of the results section is poorly structured and difficult for readers to follow. Readers cannot manage that much information. I recommend providing a brief reminder to readers of which analysis was used for each aim and/or hypothesis.What are your most important novel findings? Please explain in the first paragraph of the discussion.

Reviewers' comments:

Reviewer's Responses to Questions

**Comments to the Author**

1. Is the manuscript technically sound, and do the data support the conclusions?

Reviewer #1: Yes

Reviewer #2: No

2. Has the statistical analysis been performed appropriately and rigorously?

Reviewer #1: Yes

Reviewer #2: I Don't Know

3. Have the authors made all data underlying the findings in their manuscript fully available?

Reviewer #1: No

Reviewer #2: Yes

4. Is the manuscript presented in an intelligible fashion and written in standard English?

Reviewer #1: Yes

Reviewer #2: Yes

Reviewer #1: Thank you for allowing me to review this paper. The problem is well developed (albeit lengthily--but given the complexity of the research question and its theoretical underpinnings and merits, this is acceptable). The analytical decisions are well explained. Every strand of the findings are thoroughly discussed.

I only have one recommendation, and that is to outline multisectoral interventions/policy implications that can be derived from the study.

Reviewer #2: Acknowledgement of disparities in health is important, and this topic has potential to make a contribution to the research in this area. However, as this paper has a strong focus relating to the covid-19 pandemic, the links to general discussion/acknowledgement of health disparities and the potential impact of this work is not clear.

It would be useful to strengthen the rationale for this work – what are the wider implications of this study and its results? The relationships between parental beliefs and discussion of health disparities with young children may differ based on the type of virus (as acknowledged in some of the parent responses, their conversations may have changed due to specific concerns/risks related specifically to covid-19) and this should be acknowledged/discussed in the paper.

The methods of this paper should be expanded. Much of this section includes results and instead should describe the questionnaire development, distribution, recruitment, etc, along with a description of the analysis plan. Could the authors expand on the process for modifying existing tools/questionnaires for this questionnaire – were they pilot tested or validated for use with this study population?

In the results, several models were included and described but this was difficult to follow. It would be useful to have some descriptive statistics (e.g. baseline characteristics of the study population, response rate (if known), distribution of the response options to questions, proportion of missing responses, etc). This might help justify/explain the choice for various regression models for the reader. For the qualitative coding, why were these results included in a regression model instead of processed as qualitative data?

The discussion provided a good general overview of results, but it would be useful to review the literature on parental beliefs regarding disparities and how that impacts discussions with children, and how the results from this study may compare. As mentioned above, I believe further reflection on the implications of this work is needed.

**Do you want your identity to be public for this peer review?** For information about this choice, including consent withdrawal, please see our Privacy Policy

Reviewer #1: No

Reviewer #2: No

---

## [Author Response · Author response to Decision Letter 1]

17 Apr 2025

EDITOR’S COMMENTS

1. No information about the methods was provided in the abstract. Please provide information on the types of quantitative and qualitative data analysis used.

We have expanded the abstract to include this information.

2. Please add information about the demographics of the participants to the abstract.

This has been added as well.

3. What are your findings of each analysis? Please provide clear and concise information in the abstract.

These are now included.

4. Please avoid any promotional statements in the manuscript. (e.g., "No research has looked at.."). It is not sufficient to indicate that a research topic has not been examined before in a way that the authors will examine it.

Although we agree that this is not sufficient motivation, we do feel that it’s relevant in terms of understanding the context for this work. We have reframed such statements to be informative rather than promotional.

5. The current study section should briefly describe the research gap, aims, hypotheses/ research questions (RQs).

We have restructured the Current Study section to include this information.

6. The rationales should be provided in the literature review, not in the current study section. Avoid any new information or new citations in this section.

Thank you for this suggestion. We have moved these rationales to the literature review, and then briefly note them as relevant to the aims in the Current Study section.

7. What are your directional hypotheses? Please explain them in the "current study" section.

We originally mentioned this information in the Method section, but now clearly state that we did not have directional hypotheses in the Current Study section.

8. Which analysis was used for each hypothesis/ RQs? Justification is needed for each analysis (e.g., mixed effect logistic regression). Please see reviewers' comments.

We now explicitly state our research questions in the Current Study section and now organize the Results in terms of research questions with their corresponding analyses.

9. The current version of the results section is poorly structured and difficult for readers to follow. Readers cannot manage that much information. I recommend providing a brief reminder to readers of which analysis was used for each aim and/or hypothesis.

As stated in response to point #8 above, we have restructured the results in line with the Editor’s suggestion.

10. What are your most important novel findings? Please explain in the first paragraph of the discussion.

Thank you for this helpful suggestion. We have explained the most important novel findings in the first paragraph of the discussion.

REVIEWER 1

1. Thank you for allowing me to review this paper. The problem is well developed (albeit lengthily--but given the complexity of the research question and its theoretical underpinnings and merits, this is acceptable). The analytical decisions are well explained. Every strand of the findings are thoroughly discussed.

Thank you for the kind comments.

2. I only have one recommendation, and that is to outline multisectoral interventions/policy implications that can be derived from the study.

Although we are not familiar with multisectoral interventions, we have added a section on the implications of our work (see also responses to Reviewer 2). If the reviewer would still like us to add information on these types of interventions, we would appreciate a citation or further guidance.

REVIEWER 2

1. Acknowledgement of disparities in health is important, and this topic has potential to make a contribution to the research in this area.

Thank you for this comment.

2. However, as this paper has a strong focus relating to the covid-19 pandemic, the links to general discussion/acknowledgement of health disparities and the potential impact of this work is not clear.

We have strengthened these portions in the General Discussion (see also Reviewer 1’s suggestion about expanding on the policy implications).

3. It would be useful to strengthen the rationale for this work – what are the wider implications of this study and its results? The relationships between parental beliefs and discussion of health disparities with young children may differ based on the type of virus (as acknowledged in some of the parent responses, their conversations may have changed due to specific concerns/risks related specifically to covid-19) and this should be acknowledged/discussed in the paper.

We now expand on the rationale and wider implications of the work, as well as the potential for different results based on the type of virus (in the Discussion).

4. The methods of this paper should be expanded. Much of this section includes results and instead should describe the questionnaire development, distribution, recruitment, etc, along with a description of the analysis plan. Could the authors expand on the process for modifying existing tools/questionnaires for this questionnaire – were they pilot tested or validated for use with this study population?

We have added brief descriptions to expand on these points. We added that the disparity judgment and explanation endorsement tasks were derived from Menendez et al. (2024). We explained that the disparity judgment scale task was added to make a potentially more sensitive measure. Minor wording modifications are not uncommon in survey research, so we took discretion to alter the wording to apply each construct to the relevant social category and group. All of the individual difference measures were validated scales and were found to be reliable in the current study, according to the observed alphas (all αs > .75). Also, the Materials and method section and the Results section are separated.

5. In the results, several models were included and described but this was difficult to follow. It would be useful to have some descriptive statistics (e.g. baseline characteristics of the study population, response rate (if known), distribution of the response options to questions, proportion of missing responses, etc). This might help justify/explain the choice for various regression models for the reader. For the qualitative coding, why were these results included in a regression model instead of processed as qualitative data?

We have added a table with the descriptives for the individual difference measures, as requested. We also now note that, because we observed that the codes appeared to differ by category (age, class, race), we conducted pre-registered regression models in order to test whether these were stable differences in our sample.

6. The discussion provided a good general overview of results, but it would be useful to review the literature on parental beliefs regarding disparities and how that impacts discussions with children, and how the results from this study may compare. As mentioned above, I believe further reflection on the implications of this work is needed.

We expand on these issues in the implications section of the ms. We believe this work could inform possible interventions to help parents understand how to discuss these topics with their children.

Below is our Amended Funding Statement, as requested:

This research was supported by NSF Grant #2055164 to SAG. Any opinions, findings, and conclusions and recommendations expressed in this material are those of the authors and do not necessarily reflect the view of the National Science Foundation. There was no additional external funding received for this study. The funders had no role in study design, data collection and analysis, decision to publish, or preparation of the manuscript.

---

## [Decision Letter · Decision Letter 1]

18 May 2025

PONE-D-24-52344R1Who gets sicker and why? Parents’ perceptions of COVID-19 disparities and how they would explain them to their childrenPLOS ONE?

We have received reports from our reviewers on your manuscript, PONE-D-24-52344R1 “Who gets sicker and why? Parents’ perceptions of COVID-19 disparities and how they would explain them to their children”, submitted to PLOS ONE.

When preparing your revision, please carefully consider the reviewers’ comments, which are provided below.

We look forward to receiving your revised manuscript.

Kind regards,

Emine Ozturk, Ph.D.

Academic Editor

PLOS ONE

Reviewers' comments:

Reviewer's Responses to Questions

**Comments to the Author**

Reviewer #3: (No Response)

Reviewer #4: (No Response)

2. Is the manuscript technically sound, and do the data support the conclusions?

Reviewer #3: Yes

Reviewer #4: Partly

3. Has the statistical analysis been performed appropriately and rigorously?

Reviewer #3: Yes

Reviewer #4: Yes

4. Have the authors made all data underlying the findings in their manuscript fully available?

Reviewer #3: Yes

Reviewer #4: Yes

5. Is the manuscript presented in an intelligible fashion and written in standard English?

Reviewer #3: Yes

Reviewer #4: Yes

Reviewer #3: This is a complex, multi-faceted study that took on the enormous yet worthy task of examining how parents think about health disparities across multiple social categories and how their beliefs and motivations influence what they might discuss with their children. It appears the authors have made substantial efforts to revise the manuscript after the initial reviews. Upon my reading, a strong rationale is provided, research questions for an exploratory study are clearly outlined, and the methods are well-described. Organizing results by research question was extremely helpful to maintain focus even while addressing findings across different categories.

Overall, I recommend this article for publication with minor revisions. Below I make recommendations to clarify a few points and to expand on discussion of the findings.

Introduction: The authors provide sufficient (but not overwhelming) background information to ensure the reader has adequate context for the study design and analysis. There is an effort to equally attend to health disparities and parent socialization in relation to multiple social categories (age, race, social class) of relevance, despite disparate and often sparse bodies of literature.

1. One minor point of confusion was in the description of relevant constructs (p.8-10); it was stated that 5 constructs would be discussed, but only 4 subheadings are presented. It would help to include in the final sub-section a clarifying statement that essentialism and social constructionism are the fourth and fifth construct and explain why they are being discussed together (or to separate social constructionism into its own subheading).

Methods:

1. Under participants, given that all other demographics are listed as “self-reported,” the final statement that the sample was neither conservative nor liberal reads as an objective assessment; recommend clarifying that this is a statement of the average self-rating of participants. Conversely, it may be more concise to state early in this section that all demographic characteristics were self-reported.

2. Clarify for IMS whether there were 5 items for each of the 3 social categories (age, race, and social class) or only 5 items total. If only 5 total, creating 1 scale, why was this measure collapsed across groups whereas other measures had separate scales for each social category?

3. There remains debate about the appropriate kappa level to be deemed “reliable.” (p.19, line 387). Assessing that codes with kappa > .60 had acceptable interrater reliability would more appropriately be described as having moderate to strong/substantial agreement (e.g. McHugh, 2012; Hallgren, 2012).

Results:

1. Inconsistency in how measures are referred to and capitalized between RQs (e.g. “Disparity Judgment” p.22 and p. 26 vs “disparity judgment (forced-choice) task” p. 24). If these analyses use the same data, then would be best to clarify in RQ1 that the force-choice task data was used; if not, note it was the scale.

2. The indication of significance levels in Table 5 is extremely confusing given the text explains only specific comparisons met the corrected alpha level. Recommend providing all p-values in a separate column and using an indicator (perhaps bold font as in Table 4) to show only those that were significant at p> .017.

3. Similarly to previous comment, clarify on p. 25 line 494 that this analysis now uses the disparity judgment scale task. This allows readers to refer back to the Methods if desired to remind themselves how this measure was administered.

Discussion

1. The example provided on p.32, line 615 about protecting children from emotional harm regarding age and a grandparent seems inconsistent with the sentence and citations that follow which focus on parents’ unwillingness to discuss racial topics and that young children are aware of social inequalities. This example may fit better elsewhere.

2. The final sentence on p.33-34 does not need to be its own paragraph. Append it to the previous one or split this section into paragraphs in a logical place (perhaps line 650, “Parents’ explanation…”)

3. Some findings are summarized but not discussed. Why do the authors believe certain differences were found in light of previous literature? What are the implications of such findings?

- Why were behavioral explanations more likely when discussing social class disparities(p.34)? Knowledge about healthcare access is mentioned, but this seems to be a structural explanation. How could this relate to behavior?

- Similarly, why is the finding regarding political affiliation meaningful (p.34)? More discussion is needed regarding how this relates to past research, parent socialization practices, and/or implications for children’s development of awareness of health disparities.

-Further explanation for the social class difference related to essentialism is necessary (p.35). This seems consistent with prior research on social class beliefs in the US and that being poor is related to character rather than structural issues.

4. I appreciate the effort of the authors to address implications more thoroughly. However, it seems an acknowledgement should be made that some parents may be uninterested in learning about the existence of health disparities and that this study demonstrates that even providing established statistics did not change some parents’ minds about whether the disparity existed. The idea of an intervention focused on helping parents understand the value of “accurate explanatory frameworks” neglects the current political and social climate where what is viewed as “accurate” and factual seems up for debate. I wonder whether the authors have any suggestions for how to leverage parents’ understanding that some health disparities exist (e.g. for age) to create openness to recognizing others.

5. The paragraph on policy focuses more generally on adults and healthcare workers, but needs to find a way to circle back to parents and children. Given that parents may not be aware of (or accepting of) health disparities, an alternate way to approach this would be to consider what role other sources of socialization (educator, peers, health professionals, media) have on the development of children’s beliefs in addition to parents, particularly as children age.

Reviewer #4: The goal of this paper was to better understand whether and how parents discuss age-, status-, and race-related disparities in COVID-19 infections. The authors employed a cross-sectional design including 443 parents of a 5- to 12-year-old. Results of this exploratory study revealed age-related differences in parents’ inclinations to discuss race- and status-based disparities, and that, unsurprisingly, parents’ own attitudes and beliefs were associated with their proclivity to engage in these discussions with their children (though, the types of associated beliefs/attitudes that emerged as significant predictors were surprising). While I believe this work is important for understanding how children come to learn and understand health disparities, there remain some concerns that temper my enthusiasm. Below, I provide my specific comments and suggestions.

- Abstract: authors begin by discussing COVID health disparities and then jump to posing the question of whether parents discuss these disparities with their children. There seems to be a link missing. Perhaps open the abstract with the importance of promoting parent-child conversations about health disparities (such as covid). Authors also mention that they employed a “series of individual difference measures”. I think authors need to be a bit more specific so that the reader knows what types of measures were employed (without having to list them all).

- I was surprised to see that the paragraphs about the importance of parent-child conversations were placed so late into the introduction. I suggest moving them earlier and elaborating upon how these conversations shaped child outcomes (there’s research showing links between conversational content and youth outcomes during the pandemic).

- The connections between authors’ first aims and the aim regarding individual differences in attitudes weren’t well streamlined. The attitudes/beliefs authors explore aren’t integrated within a theoretical framework, which makes it difficult to understand why authors selected all of these (i.e., how are these attitudes and beliefs differentiated and how do they conceptually relate to one another? How does it meaningfully contribute to authors’ overarching narrative). I urge authors to think about which attitudes/beliefs are core to their investigation and perhaps remove or move analyses concerning others to the supplementary material.

- I was surprised that authors did not have any specific hypotheses, especially for aim 4 as there is some research on this topic already (at least in the domain of race-based conversations)

- Authors reported a sensitivity analysis in g*power but it sounds like they conducted an a priori power analysis to determine their sample size. Authors are also missing information about total predictors, tested predictors, power value, and alpha value.

- Do authors know the States from which this data was primarily collected? It’s great that authors collected data on political affiliation—it definitely matters for what parents talk about. I recommend including political leaning as a covariate in all multivariate analyses, as well as parent and/or child gender and age (if not already a predictor)

- Why didn’t authors explore race-related differences in their outcomes? 39% of the sample was bipoc—there may be some interesting differences in authors’ findings, especially the thematic analysis of content, strategies, and reasons parents provided.

- The parent explanation section is very interesting and could stand alone as a separate paper if authors were to further describe the findings qualitatively. Authors could unpack parents’ explanations further, especially regarding differences in themes by key demographics to determine who is saying what and how they are discussing the content. This would have important intervention implications.

- More information about the coding process is needed (i.e., how codes were developed and revised). Authors mention basing their coding scheme on Menendez et al, but it’s not clear whether the codebook was adapted for use with this data. Employing appropriate thematic analysis reporting procedures is recommended (Terry & Hayfield, 2021 could be helpful)

- I’m unclear about what the purpose of the confidence ratings was.

- RQ4: authors need to provide bivariate correlations between their attitudes/beliefs measures. The findings are quite perplexing. The results might be easier to digest if presented in a table rather than in text. The descriptives could be moved to the supplementary material. Was multicollinearity a factor if all variables were included in the same models?

- The political affiliation findings/discussion seem out of place.

- By the end of the paper, I am left wondering about the novelty of the findings. Authors present many results, but I think they could work toward crafting a more cohesive narrative to tie all the findings together. I think authors can leverage their findings in the implications section better.

- There are a few typos throughout.

**Do you want your identity to be public for this peer review?** For information about this choice, including consent withdrawal, please see our Privacy Policy

Reviewer #3: No

Reviewer #4: No

---

## [Author Response · Author response to Decision Letter 2]

2 Jul 2025

PONE-D-24-52344R1

Reviewer 3

This is a complex, multi-faceted study that took on the enormous yet worthy task of examining how parents think about health disparities across multiple social categories and how their beliefs and motivations influence what they might discuss with their children. It appears the authors have made substantial efforts to revise the manuscript after the initial reviews. Upon my reading, a strong rationale is provided, research questions for an exploratory study are clearly outlined, and the methods are well-described. Organizing results by research question was extremely helpful to maintain focus even while addressing findings across different categories. Overall, I recommend this article for publication with minor revisions. Below I make recommendations to clarify a few points and to expand on discussion of the findings.

Thank you for the positive comments.

Introduction: The authors provide sufficient (but not overwhelming) background information to ensure the reader has adequate context for the study design and analysis. There is an effort to equally attend to health disparities and parent socialization in relation to multiple social categories (age, race, social class) of relevance, despite disparate and often sparse bodies of literature.

1. One minor point of confusion was in the description of relevant constructs (p.8-10); it was stated that 5 constructs would be discussed, but only 4 subheadings are presented. It would help to include in the final sub-section a clarifying statement that essentialism and social constructionism are the fourth and fifth construct and explain why they are being discussed together (or to separate social constructionism into its own subheading).

Thank you for your suggestion. We have separated essentialism and social constructionism by subheading to clarify that they are conceptually distinct constructs.

Methods:

1. Under participants, given that all other demographics are listed as “self-reported,” the final statement that the sample was neither conservative nor liberal reads as an objective assessment; recommend clarifying that this is a statement of the average self-rating of participants. Conversely, it may be more concise to state early in this section that all demographic characteristics were self-reported.

We added a brief statement indicating that all demographics were self-reported.

2. Clarify for IMS whether there were 5 items for each of the 3 social categories (age, race, and social class) or only 5 items total. If only 5 total, creating 1 scale, why was this measure collapsed across groups whereas other measures had separate scales for each social category?

The measure was 5 items total, and we have clarified this in the Method section. The reason we did not have separate questions for each social category was that we did not want to presuppose which category (out of the pair) participants were more prejudiced about. For example, some participants may have more prejudice against rich people than poor people, and others may have had the reverse. None of the other scales that we had involved picking just one of the two categories in a pair. What we most cared about was whether participants were motivated not to show prejudice broadly; therefore, the single 5-item scale was slightly modified to include general wording.

3. There remains debate about the appropriate kappa level to be deemed “reliable.” (p.19, line 387). Assessing that codes with kappa > .60 had acceptable interrater reliability would more appropriately be described as having moderate to strong/substantial agreement (e.g. McHugh, 2012; Hallgren, 2012).

We now acknowledge this in our qualitative coding section.

Results:

1. Inconsistency in how measures are referred to and capitalized between RQs (e.g. “Disparity Judgment” p.22 and p. 26 vs “disparity judgment (forced-choice) task” p. 24). If these analyses use the same data, then would be best to clarify in RQ1 that the force-choice task data was used; if not, note it was the scale.

We apologize for the inconsistencies, which have been corrected.

2. The indication of significance levels in Table 5 is extremely confusing given the text explains only specific comparisons met the corrected alpha level. Recommend providing all p-values in a separate column and using an indicator (perhaps bold font as in Table 4) to show only those that were significant at p> .017.

We have clarified that all selections for the disparity judgment (forced-choice) task were significantly different from chance (33.33%), noting that some were below chance and others were above chance. We also indicate that all comparisons to chance were p < .001.

3. Similarly to previous comment, clarify on p. 25 line 494 that this analysis now uses the disparity judgment scale task. This allows readers to refer back to the Methods if desired to remind themselves how this measure was administered.

We have briefly added that these analyses were for the disparity judgment (scale) task.

Discussion:

1. The example provided on p.32, line 615 about protecting children from emotional harm regarding age and a grandparent seems inconsistent with the sentence and citations that follow which focus on parents’ unwillingness to discuss racial topics and that young children are aware of social inequalities. This example may fit better elsewhere.

We appreciate the suggestion, and have reorganized the paragraphs in this section so that the examples and citations are more appropriate for the points that are made.

2. The final sentence on p.33-34 does not need to be its own paragraph. Append it to the previous one or split this section into paragraphs in a logical place (perhaps line 650, “Parents’ explanation…”)

We have added that final sentence to the prior paragraph.

3. Some findings are summarized but not discussed. Why do the authors believe certain differences were found in light of previous literature? What are the implications of such findings?

We have added to the Discussion, adding interpretations and implications for findings that had been only summarized.

- Why were behavioral explanations more likely when discussing social class disparities(p.34)? Knowledge about healthcare access is mentioned, but this seems to be a structural explanation. How could this relate to behavior?

To clarify: it is not that behavioral explanations were more likely for social class disparities (rather, as seen in Table 8, structural explanations were more likely). Instead, the finding discussed on p. 34 was that parents who provided behavioral explanations were more likely to say they would discuss these disparities with their children. We suspect that the reviewer is correct, that behavioral explanations may often be assumed to be the result of structural factors. Importantly, as noted, parents who provided structural explanations were also more likely to say they would discuss these disparities with their children. We have edited this paragraph to make these findings clearer.

- Similarly, why is the finding regarding political affiliation meaningful (p.34)? More discussion is needed regarding how this relates to past research, parent socialization practices, and/or implications for children’s development of awareness of health disparities.

We have elaborated on this finding and how it relates to past research.

- Further explanation for the social class difference related to essentialism is necessary (p.35). This seems consistent with prior research on social class beliefs in the US and that being poor is related to character rather than structural issues.

We have expanded our interpretation regarding this finding in the Discussion section.

4. I appreciate the effort of the authors to address implications more thoroughly. However, it seems an acknowledgement should be made that some parents may be uninterested in learning about the existence of health disparities and that this study demonstrates that even providing established statistics did not change some parents’ minds about whether the disparity existed. The idea of an intervention focused on helping parents understand the value of “accurate explanatory frameworks” neglects the current political and social climate where what is viewed as “accurate” and factual seems up for debate. I wonder whether the authors have any suggestions for how to leverage parents’ understanding that some health disparities exist (e.g. for age) to create openness to recognizing others.

We have added an acknowledgement that our results may indicate parents’ disinterest in learning about social disparities in the Discussion. We now also acknowledge the role of the politicalization of COVID-19 policies and health recommendations, and suggest that in future research it would be of interest to determine how parents reason about other health disparities that may not be subject to the same level of politicalization as COVID-19 (e.g., the flu). We seriously considered the reviewer’s interesting suggestion of leveraging parents’ understanding of (e.g.) age disparities, as a foundation for introducing other disparities. In the end, however, we opted not to add this suggestion to the ms., as we are concerned that doing so could encourage parents to treat other disparities (e.g., race, social class) as biological, much as they do for age.

5. The paragraph on policy focuses more generally on adults and healthcare workers, but needs to find a way to circle back to parents and children. Given that parents may not be aware of (or accepting of) health disparities, an alternate way to approach this would be to consider what role other sources of socialization (educator, peers, health professionals, media) have on the development of children’s beliefs in addition to parents, particularly as children age.

We have incorporated some discussion on potential ways that other socialization agents (i.e., educators, health professionals) can contribute to children’s developing beliefs of health disparities.

Reviewer 4

1. Abstract: authors begin by discussing COVID health disparities and then jump to posing the question of whether parents discuss these disparities with their children. There seems to be a link missing. Perhaps open the abstract with the importance of promoting parent-child conversations about health disparities (such as covid). Authors also mention that they employed a “series of individual difference measures”. I think authors need to be a bit more specific so that the reader knows what types of measures were employed (without having to list them all).

We made these edits.

2. I was surprised to see that the paragraphs about the importance of parent-child conversations were placed so late into the introduction. I suggest moving them earlier and elaborating upon how these conversations shaped child outcomes (there’s research showing links between conversational content and youth outcomes during the pandemic).

We have adjusted parts of the Introduction and added recent research findings emphasizing the role of parent-child conversations on children’s social and conceptual development in the COVID-19 context.

3. The connections between authors’ first aims and the aim regarding individual differences in attitudes weren’t well streamlined. The attitudes/beliefs authors explore aren’t integrated within a theoretical framework, which makes it difficult to understand why authors selected all of these (i.e., how are these attitudes and beliefs differentiated and how do they conceptually relate to one another? How does it meaningfully contribute to authors’ overarching narrative). I urge authors to think about which attitudes/beliefs are core to their investigation and perhaps remove or move analyses concerning others to the supplementary material.

Given that these measures were pre-registered, we do not feel comfortable excluding them from the manuscript. However, we note that we took an exploratory approach, given little prior work addressing these questions. We also added our rationale for selecting these measures, namely, they have been found to relate to how people think about social categories and/or parenting behaviors associated with conversations about social categories.

4. I was surprised that authors did not have any specific hypotheses, especially for aim 4 as there is some research on this topic already (at least in the domain of race-based conversations)

Given the exploratory nature of Research Question 4, we did not preregister any hypotheses. For this reason, we do not add any post-hoc hypotheses for this research question.

5. Authors reported a sensitivity analysis in g*power but it sounds like they conducted an a priori power analysis to determine their sample size. Authors are also missing information about total predictors, tested predictors, power value, and alpha value.

The reviewer is correct that we did an a priori power analysis, and we have made changes to the text to reflect this. We have also added the missing pieces of information that the reviewer requested to the manuscript.

6. Do authors know the States from which this data was primarily collected? It’s great that authors collected data on political affiliation—it definitely matters for what parents talk about. I recommend including political leaning as a covariate in all multivariate analyses, as well as parent and/or child gender and age (if not already a predictor)

We had at least one participant from each of 46 states and the District of Columbia, with the top three most common states being New York (n=36), Texas (n=36), and Florida (n=32). The states not represented in our data are Hawaii, Montana, Vermont, and Wyoming. We mention this in the Method. Regarding the covariates mentioned, we preregistered having the participants’ child’s age in our discussion likelihood models. We did not ask participants to indicate the gender of their child. Finally, we did not preregister including political leaning in our models. However, we conducted exploratory analyses predicting parents’ likelihood of discussing the disparities based on their highest educational attainment, political leaning, and political affiliation in three separate regression models, one for each disparity. We did not find political leaning to be a significant factor for discussing any of the three disparities, even when controlling for the socio-psychological individual differences.

7. Why didn’t authors explore race-related differences in their outcomes? 39% of the sample was bipoc—there may be some interesting differences in authors’ findings, especially the thematic analysis of content, strategies, and reasons parents provided.

The reviewer is correct that our sample consisted of a relatively large proportion of non-White participants. However, because our study’s design focused on Black-White differences, we do not believe it would be conceptually appropriate to collapse over all non-White participants, as we did not include information about disparities between other ethnic-racial groups. In our Limitations section, we advocate for future research to include more diverse samples to obtain perspectives of more individuals from groups who were most affected. Additionally, our data will be publicly available for anyone interested in conducting these analyses.

8. The parent explanation section is very interesting and could stand alone as a separate paper if authors were to further describe the findings qualitatively. Authors could unpack parents’ explanations further, especially regarding differences in themes by key demographics to determine who is saying what and how they are discussing the content. This would have important intervention implications.

Thank you for this suggestion. For the present paper, we will keep the parent explanation section succinct, but your idea is an interesting one that we can consider further in the future.

9. More information about the coding process is needed (i.e., how codes were developed and revised). Authors mention basing their coding scheme on Menendez et al, but it’s not clear whether the codebook was adapted for use with this data. Employing appropriate thematic analysis reporting procedures is recommen

---

## [Decision Letter · Decision Letter 2]

27 Aug 2025

Who gets sicker and why? Parents’ perceptions of COVID-19 disparities and how they would explain them to their children

PONE-D-24-52344R2

Dear Dr. Menendez,

We’re pleased to inform you that your manuscript has been judged scientifically suitable for publication and will be formally accepted for publication once it meets all outstanding technical requirements.

Kind regards,

Janet E Rosenbaum, Ph.D.

Academic Editor

PLOS ONE

Additional Editor Comments (optional):

Reviewers' comments:

Reviewer's Responses to Questions

**Comments to the Author**

Reviewer #3: All comments have been addressed

Reviewer #4: (No Response)

2. Is the manuscript technically sound, and do the data support the conclusions?

Reviewer #3: Yes

Reviewer #4: (No Response)

3. Has the statistical analysis been performed appropriately and rigorously?

Reviewer #3: Yes

Reviewer #4: (No Response)

4. Have the authors made all data underlying the findings in their manuscript fully available?

Reviewer #3: Yes

Reviewer #4: (No Response)

5. Is the manuscript presented in an intelligible fashion and written in standard English?

Reviewer #3: Yes

Reviewer #4: (No Response)

Reviewer #3: (No Response)

Reviewer #4: I thank the authors for reviewing their work and considering some of my comments. I provide additional feedback below. My remaining concerns are mostly regarding flow, theoretical integration, and density of findings. Overall, I believe this paper provides a meaningful contribution to the field.

Overarching comment:

- Authors argue for maintaining their pre-registered ideas throughout the response letter. The purpose of pre-registration is to maintain scientific transparency and having a preregistration does mean that authors cannot further improve and adapt the narrative of their paper (so long as critical deviations are reported). Here are a few papers that delineate helpful uses of preregistrations that do not contradict the natural development of authors’ ideas and are in line with best scientific practices:

https://myscp.onlinelibrary.wiley.com/doi/10.1002/jcpy.1209

https://journals-sagepub-com.proxy.lib.sfu.ca/doi/full/10.1177/25152459231213802

https://online.ucpress.edu/collabra/article/10/1/117094/200749/When-and-How-to-Deviate-From-a-Preregistration

Prior comments:

- Abstract: Authors are still lacking flow from one point to another. E.g., the second sentence addresses gaps in the literature and the following statement states how research questions will be addressed. I recommend changing the second sentence to be explicitly about authors’ research questions or adjusting the language. Examining associations between parents’ attitudes and beliefs about social categories and their relation to conversations is a core focus, but this is not posed as a research question. Please clearly outline specifically what the aims/questions are within the abstract to increase clarity (how authors state their aims at the end of the introduction section on page 5 is great).

- “We first want to clarify that we did not employ a full thematic analysis, but thank the reviewer for guiding us on how to appropriately report our qualitative analyses. We employed a codebook approach with a combination of inductive and deductive codes. We have added information in the Method section about the process of creating the coding scheme, and provide details on which codes came from prior codebooks established in the literature and which were created specifically for this paper.”

o Thank you for clarifying. There are different coding approaches that fall under the umbrella of thematic analysis that nevertheless require appropriate reporting. The authors’ revisions better describe their process and the coding scheme.

- “Given that these measures were pre-registered, we do not feel comfortable excluding them from the manuscript. However, we note that we took an exploratory approach, given little prior work addressing these questions. We also added our rationale for selecting these measures, namely, they have been found to relate to how people think about social categories and/or parenting behaviors associated with conversations about social categories.”

o Authors still lack meaningful integration. Why these five measures? Do they each tell us something unique and meaningful about certain aspects of individual differences and how they relate to parent-child communication? Some more coherence beyond indicating that the measures were included in the preregistration (see overarching comment above) and related to how people think about social categories is needed.

- “We created a new table to display our Q4 results in a more digestible format. We added the descriptives and bivariate correlations to the Supporting Material. Additionally, multicollinearity was not an issue for any of the models, with all variance inflation factors below 5.05.”

o A typically recommended VIF cutoff is 5.

Additional comments:

- Authors would benefit from adjusting their subheadings to be more specific (e.g., “Parental input regarding race, social class, and age”  “Parent-child communication about race, social class, and age”; Attitudes and beliefs about social categories  The Role of Parental attitudes and beliefs about social categories on communication

- It would be helpful if authors could separate the procedure from the measure descriptions for clarity and flow. Perhaps a flow chart of the procedure would benefit the clarity of the paper.

- Research on intergroup helping by Jellie Sierksma shows that children tend to help outgroup members more when they perceive them as less competent. This could resonate well with your findings re: race disparities and discussions.

- Tables require APA formatting

**Do you want your identity to be public for this peer review?** For information about this choice, including consent withdrawal, please see our Privacy Policy

Reviewer #3: No

Reviewer #4: No

---

## [Editor Report · Acceptance letter]

PONE-D-24-52344R2

PLOS ONE

Dear Dr. Menendez,

I'm pleased to inform you that your manuscript has been deemed suitable for publication in PLOS ONE. Congratulations! Your manuscript is now being handed over to our production team.

Kind regards,

on behalf of

Dr. Janet E Rosenbaum

Academic Editor

PLOS ONE